# **Bias Correction of Satellite-Based Rainfall Estimates for Modeling Flash Floods in Semi-Arid regions: Application to Karpuz River, Turkey**

Mohamed Saber <sup>1, 2, 3</sup>, Koray K. Yilmaz<sup>2</sup>

<sup>1</sup>Geology Department, Faculty of Science, Assiut University, Assiut 71516, Egypt
 <sup>2</sup>Department of Geological Engineering, Middle East Technical University, 06800, Ankara, Turkey
 <sup>3</sup>Water Resources Research Center, DPRI, Kyoto University, Goka-sho, Uji City, Kyoto 611-0011, Japan;

Correspondence to: Mohamed Saber (mohamedmd.saber.3u@kyoto-u.ac.jp)

Abstract: This study investigates the utility of gauge-corrected satellite-based rainfall estimates in simulating flash floods at

10 Karpuz River - a semi-arid basin in Turkey. Global Satellite Mapping of Precipitation (GSMaP) product was evaluated with the rain gauge network at monthly and daily time-scales considering various time periods and rainfall rate thresholds. Statistical analysis indicated that GSMaP shows acceptable linear correlation coefficient with rain gauges however suffers from significant underestimation bias. A rainfall rate threshold of 1 mm/month was the best choice to improve the match between GSMaP and rain gauges implying that appropriate threshold selection is critically important for the bias correction.
Multiplicative bias correction was applied to GSMaP data using the bias factors calculated between GSMaP and observed rainfall. Hydrological River Basin Environmental Assessment Model (Hydro-BEAM) was used to simulate flash floods at the hourly time scale driven by the corrected GSMaP rainfall data. The model parameters were calibrated for flash flood events

during October-December 2007 and then validated for flash flood events during October-December 2009. The results show that the simulated surface runoff hydrographs reasonably coincide with the observed hydrographs.

**Keywords:** Flash floods modeling; semi-arid regions; bias correction; GSMaP; Antalya; Turkey

#### 1. Introduction

Spatio-temporal variation of rainfall is important to understand the hydrological and climatic characteristics of watersheds, as well as for planning effective water management and hazard mitigation strategies for water-related disasters such as flash floods and droughts. The influence of rainfall representation on the modelling of the hydrologic response is expected to depend on complex interactions between the rainfall space-time variability, the variability of the catchment soil and landscape properties, and the spatial scale (i.e. catchment area) of the problem (Obled et al., 1994; Woods and Sivapalan, 1999; Bell and Moore, 2000; Smith et al., 2004). River hydrograph forecasts are highly dependent upon the input rainfall data used. Streamflow in arid and semi-arid regions is characterized by rapid response to intense rainfall events. Such events frequently have a high degree of spatial variability. Coupled with poorly gauged rainfall data this situation sets a fundamental limit on the capacity of the rainfall-runoff models to reproduce the observed flow (Wheater et al., 2008), thus hampering prediction

efforts. It has been widely stated that the major limitation of the development of arid-zone hydrology is the lack of high quality observations (McMahon and Greene, 1979; Pilgrim et al., 1988).

Accurate temporal knowledge of global precipitation is definitely important for understanding the multi-scale interactions among weather, climate and ecological systems, as well as for improving the ability to manage the available water resources

- and predict high-impact weather events including hurricanes, floods, droughts and landslides (Hou et al., 2008). Rain gauge observations yield relatively accurate point measurements of precipitation but are not well distributed and not available over most oceanic and unpopulated land areas (Xie and Arkin, 1996; Petty and Krajewski, 1996). In particular, the arid and semiarid regions are suffering from poor rain gauge network coverage and the lack of continuous observations, especially at the hourly timescale, which will be consequently a challenge for the flash floods mitigation efforts. Therefore, algorithms developed for bias correction of the satellite-based rainfall data are desperately needed in such regions.

The combination of satellite measurements and gauge data is in great need to enhance spatio-temporal rainfall estimation (Chiu et al., 2006a). Global satellite-based rainfall datasets are becoming increasingly available in different spatial and temporal resolutions and freely accessible. For example, widely used global rainfall datasets include; the Global Satellite Mapping of

10

Precipitation (GSMaP) (Okamoto et al., 2005); (Ushio et al., 2009), Precipitation Estimation from Remotely Sensed Information using Artificial Neural Networks (PERSIANN) (Hsu et al., 1999; Sorooshian et al., 2000), Climate Prediction Center (CPC) Morphing technique (CMORPH) (Joyce et al., 2004; Xie, 2013) Tropical Rainfall Measuring Mission (TRMM) Multi-satellite Precipitation Analysis (TMPA) (Huffman et al., 2010).

- 5 Recent decades are marked by increasing flash flood hazards in many regions over the world due to increase in the frequency of flash floods as well as rapid urbanization. The term 'flash flood' identifies a rapid hydrological response, with water levels reaching a peak within less than one hour to a few hours after the onset of the generating rain event (Creutin et al., 2013; Collier, 2007; Younis et al., 2008). The most important challenge to model implementation and calibration to simulate flash floods in semi-arid and arid regions is the lack of necessary observational networks of both rainfall and discharge.
- slope, impermeable surfaces, and sudden release of impounded water over small basins (Georgakakos, 1986). Flood occurrences are complex since they depend on interactions between many geological and morphological characteristics of the basins, including rock types, elevation, slope, sediment transport, and flood plain area. Moreover, hydrological phenomena, such as rainfall, runoff, evaporation, and surface and groundwater storage (Farquharson et al., 1992; Flerchinger

Responsible factors for the short duration of the flash flood include intense rains that persist on an area for a few hours, steep

15and Cooley, 2000; Nouh, 2006) affect floods. According to (Few et al., 2004), each flood acquires some particular and inherent<br/>characteristics of the occurrence locality, such as flow velocity and height, duration, and rate of water-level rise.

In many countries and regions of the world, flash floods are the most costly natural hazards in terms of both loss of human lives and material damage (Fattorelli et al., 1999; Creutin et al., 2013). In particular, arid and semi-arid regions have become more vulnerable to flash floods than before possibly due to the global climate change and rapid population growth. The main

20 obstacle to study flash floods is clearly the lack of reliable observations in most of the flash flood prone basins. Furthermore, the danger also comes from the rarity of the phenomenon, which demands new observation strategies, as well as new forecasting methodologies.

5

Various problems associated with forecasting flash floods caused by convective storms over semi-arid basins have been studied by (Michaud and Sorooshian, 1994). Rapidly increasing availability of good quality weather radar observations is greatly expanding our ability to measure and monitor rainfall distribution at the space and time scales which characterize the flashflood events (Borga et al., 2007). Moreover, some hydrological approaches and understanding of runoff characteristics in arid environment have been developed by (Saber et al., 2010b; Saber et al., 2013), and a method for estimating flash flood peak discharge, hydrograph, and volume has been presented by (Koutroulis and Tsanis, 2010). Hence an integrated and comprehensive research regarding flash flood modeling and forecasting approaches as well as mitigation strategies are desperately needed in semi-arid and arid regions.

A suite of models is available to represent rainfall-runoff relationships, but they have limitations in the hydrologic parameters that are used to describe the rainfall–runoff process in semi-arid and arid systems (Wheater et al., 1993). It has been widely stated that the major limitation of the development of arid-zone hydrology is the lack of high quality observations (McMahon and Greene, 1979; Pilgrim et al., 1988). Thus, implementation of a hydrological model driven by locally corrected satellitebased rainfall estimates could be useful in overcoming majority of the problems in simulating flash floods, thus potentially serve as a tool for hazard mitigation and sustainable development of the target basin.

15 A comparative study has been done (Saber, 2010; Saber et al., 2013) between GSMaP and the Global Precipitation Climatology Center (GPCC) in the arid and semi-arid regions over the globe to characterize the GSMaP product bias as compared to observed GPCC product. Additional analysis using the local raingauge network at a semi-arid region will be considered in this research for further validation of GSMaP product. Consequently, the forecasting models driven by the bias-corrected satellitebased rainfall datasets are expected to be more powerful and reliable. This study aims to compare GSMaP product with the gauge-based precipitation estimates in Karpuz River located in Antalya, Turkey in an effort to devise a correction methodology for the GSMaP product to drive a hydrological model for flash-flood simulation. Due to rapid occurrence of flash floods at sub-daily time scales generally hourly spatio-temporal rainfall data is used for flash flood simulation studies. Thus, satellite-

based rainfall data at the hourly timescale with continuous availability in time provides an alternative to ground-based

observations for flash flood simulation studies. The paper consists of two main parts. First, data comparison and the procedure for correction of Satellite-based rainfall data (GSMaP) is introduced. Second, flash flood modeling through a hydrological model with calibrated and validated parameters for Karpuz River - a semi-arid basin in Antalya, Turkey is provided.

#### 2 Study Area and Datasets

5 The study area is the Karpuz River Basin located to the west of the city of Antalya situated in the Mediterranean region of Turkey. The study area lies between 30.50E-32.50E longitude bands and 36.00N-37.50N Latitude bands with a total area of about 1920 km2, where the land data pixels were only considered for the analysis (Fig. 1).