# Peer review of "Bias Correction of Satellite-Based Rainfall Estimates for Modeling Flash Floods in Semi-Arid regions: Application to Karpuz River, Turkey"

_Natural Hazards and Earth System Sciences, 2016_

## Referee Comment (RC1) · Anonymous Referee #1 · 19 Dec 2016

GENERAL COMMENTS

This study investigates the utility of gauge-corrected satellite-based rainfall estimates in simulating flash floods at Karpuz River - a semi-arid basin in Turkey. Global Satellite Mapping of Precipitation (GSMaP) product was evaluated with the rain gauge network at monthly and daily time-scales considering various time periods and rainfall rate thresholds. Literature lacks of studies using satellite rainfall estimates for flash flood modelling therefore the paper is relevant and of interest for the readers of the journal. Despite this, I think the paper contains serious shortcomings and its presen-

Interactive
comment

tation is very poor. The main drawback is the analysis of the rainfall which is carried out on a monthly basis while flash flood occurrence time scale is often sub-daily and even sub-hourly (in this respect the authors claim in the abstract that the analysis has been carried out at daily time steps but no daily results can be found in the paper). An analysis at hourly and daily rainfall would be more appropriate for the study. The interpolation of rain gauges on the GSMaP grid seems wrong (I see many artefacts in Figures 15 and 22). An adequate discussion of the potential error of the interpolation (5 stations for obtaining rainfall at 0.1*0.1 degree on 2 x 1.5 degree area) should be present in the manuscript.

The presentation of the paper lacks of an appropriate organization: 1.Intro section rationale should be:

A. Flash flood problems.

B. Use of gauge rainfall network problematic because there are too little number of stations

C. Possible alternative use of satellite data, problem with bias with satellite data,

D. Bias correction improves the hydrological model.

In the way it is presented it is difficult to follow.

2. Datasets description is totally missing (GSMaP is not described at all) and study area is described twice in section 2 and in section 4.1. after the result section. In section 4.1. no further information is given about the catchment characteristics neither about the discharge time series, event selection and so on. Here, only additional info about the flash flood problem are given (material that fits more for the introduction section).

3. The number of figures is enormous and redundant. Tables often contain the same information of the figures. 4. Performance scores are inadequate. NSE is used in rainfall assessment and not in flood assessment. I think it would be interesting to
use categorical performance scores (Probability of detection and False alarm ratio) for rainfall assessment and RMSE and use NSE in the flood part.

Based on that, I suggest the paper to be not acceptable and suggest to resubmit after being improved.

I also have other comments that I will list below in order of appearance in the manuscript indicating also their relevance. The authors could take them into account for improving the manuscript.

MODERATE: Pag. 3 lines 17-22 – Pag 4 lines 1-8. This part should be moved at the beginning of the manuscript.

MODERATE: Pag. 2 lines 19-20. It seems the sentence is not a consequence of what is written before. Consider moving after describing potential problems of bias in satellite rainfall estimates.

MINOR: Figure 1. Merge this figure with Figure 16.

MAJOR: Pag. 4 lines 21-22. Why do you assess rainfall at monthly time scale then?

MAJOR. Pag 6 line 1- GSMap. Is not described in the text. Its description is relevant for the paper.

MODERATE: Section 3. Explain better the difference between the PBIAS and BIAS and what information they should give one with respect to the other.

MODERATE: Pag. 8 lines 17-20. Not clear.

MINOR: figure 10 contains the same information of table 3. Consider removing.

MAJOR. Section 3.1. point vs. grid comparison. Is not described in the methodology. What is the objective of this analysis?

MODERATE. Pag 16 line 1-10. Please try to describe better this part. It seems very important for the paper.

MAJOR. Figure 15(a) distribution of rainfall is very strange. Please check.

MAJOR. Remove this section and merge with section 2.

MAJOR. Pag 21 lines 13-14. "various remotely ... model". Which are the other remotely sensed datasets used in the study?

MODERATE. Table 7. R=0.6 would probably mean NSE <0.5. For flash flood analysis this is not a good performance score. Please discuss and add NSE in the tables.

---

## Referee Comment (RC2) · Anonymous Referee #2 · 2 Jan 2017

The work presents a study on the application of satellite-rainfall data for flash flood simulations for a basin in the Mediterranean region of Turkey. The first part of the work focuses on the analysis and bias correction of satellite rainfall estimates with reference to available rain gauges. The second part presents the hydrologic simulations based on corrected satellite-rainfall estimates. The topic of satellite-based flood simulation is of interest, particularly for data scarce regions as the one considered in this study. However, the work presented in this manuscript is limited in many aspects, which I highlight in more detail below, thus making it unsuitable for publication.

[Figure]

Major comments

1. Clarity and originality of objectives. It is not clear which is the major added value of this study. Is this the first work regarding satellite-rainfall and flash floods? I do not think so. Just a simple google search using the keywords (satellite, rainfall, flash floods) return several other past and recent works, which in fact were completely ignored from the authors. References need to be updated and authors need to contrast their objectives, methods and findings to what is already shown from previous studies. This will help the reader (and reviewer) to understand what is the added value of this work and subsequently how significant this is.

2. Structure and coherence of manuscript. The coherence and structure of text in current version of the manuscript is very poor. At a large part I had the feeling that I was reading a document that was put together by simply pasting sections from a master thesis or similar. Many statements are unnecessarily repeated. The flow in developing the context in the introduction, but also other parts, is incomplete and sometimes certain statements are out of context. I provide some examples in the list of specific comments below.

3. Limitations in methodology and analysis. There is a general lack of clarity in many steps of the methodology that limits the understanding (at least of this reviewer). For example, was the bias adjustment factor derived and applied at monthly scale? Why not at daily scale? Also, was the spatial resolution of the hydrologic model at 1km? Do you think this is appropriate for modeling flash floods? What is the rational for selecting a rainfall threshold for the analysis? This is not explained well in the manuscript. Is the bias correction applied also for the values below the selected threshold (i.e. you may be increasing the "noise" as well if you do that). I provide more specific comments below.

4. Limitations in findings and conclusions. According to the conclusions of the authors one of the important findings relate to the selection of an appropriate threshold for

the correction of satellite. What is the rational for selecting this threshold? What are the potential dependencies of this threshold? This is important to generalize findings. But most importantly, several satellite-correction techniques have been proposed over the last decade that involve more advanced approaches (e.g. stochastic error models, distribution matching procedures etc) than the one proposed. Is the simple method applied in this study superior? If the authors wanted to demonstrate simply the effect of a simple correction on the hydrologic simulations, then they should have at least present the hydrologic analysis for both corrected and not corrected results.

The authors state also that the "bias factors calculated in this study could be used for hydrological applications at any region with the same climatic conditions." Given the large differences of satellite-rainfall error properties reported from numerous evaluation studies, I am considering naïve to state that an adjustment factor could be unconditionally applied to another region of same climate...

Overall, I believe that more work is needed to bring the manuscript to a level adequate of publication and the work should focus on advancing the analysis as well as the writing/presentation of results.

Specific comments

1. L13-14 in the abstract is not clear. Please revise. In general the abstract is too descriptive and does not provide the overall objective of this work clearly. 2. P2, L4: "hydrologic response is expected....". It is not expected, it is proven by several studies. 3. P3, L1-4: Why you mention all these products when you are just using 1 of them? And by the way, why you considered only 1 product and did not include at least the high-res CMORPH? 4. P3,L8: "...model implementation and calibration..." please revise it is not clear. 5. P3,L15-16: "..each flood acquires some particular and inherent characteristics of the occurrence locality". The meaning is totally unclear here. 6. P4, L9-10: You mention the issue of model parameter uncertainty but how that relates to the rest of the context discussed in this work? 7. P8, L11: "underestimated

bias" is not appropriate wording. Rainfall is underestimated not bias. 8. P8, L18-19: Interpretation provided is not meaningful. A rational for the selection of 1mm threshold is needed. 9. Why you present results on monthly scale (e.g. Fig2)? This temporal scale is completely our of flash flood context. 10. What is the difference between fig10 and fig 11? 11. P16,L16: "...tendency to underestimate rainfall..." is a result of NO DETECTION which is different from underestimation. 12. P17,L4-8: Is this the first time that these findings are reported? 13. Section 3.2: Why repeating the procedure? Is bias adjustment factor from monthly comparison applied at hourly scale? 14. Fig17 is not necessary as it is not related to main topic of this work. It is enough to provide reference to Ogzuler, 2003. 15. P21, L7: I am not sure that "physically-based" is an appropriate description for this model. Do you consider the curve number and linear storage models as "physically-based" approaches? 16. Fig19. "disaggregation (1km)". There was no mention on disaggregation of satellite rainfall estimates. If you mean that GSMaP rainfall was simply mapped on 1-km model pixels then this is not disaggregation. 17. P25,L1: satellite rainfall data offer better "coverage" not "spatial resolution" as mentioned here 18. P28, L1-2: Explain how you identify areas prone to flash flood events? 19. Fig.21 is simply a representation of the river network and in fact a coarse representation of the river network. It is not clear to me how this info can be used for "water management and disaster rist reduction" as stated by the authors. 20. P29, L1-2: "The main objective of the study is to enhance the capability of flash flood simulation using the corrected satellite-based rainfall data sets". You need to provide a comparison of hydrologic simulations between uncorrected and corrected. Also you hydrologic analysis should be focused on the flash flood events by providing error properties on their corresponding characteristics (e.g. flood peak estimation) instead of an overall efficiency metric that includes the whole time series.
* * *

---

## Author Comment (AC1) · 10 Feb 2017

Dear Editor,

Please find below our detailed responses to the reviews made by Reviewer #1. We found the Reviewer comments very helpful and believe that the revisions detailed below will represent a significant improvement over the original submission. In our responses below, we have clearly noted each comment and our specific response to those comments so that you can make a well-informed judgment. We would like to thank Reviewer #1 for his/her detailed comments on the paper.

Sincerely,

Mohamed Saber and Koray K. Yilmaz

GENERAL COMMENTS:

**Overview Comment:** This study investigates the utility of gauge-corrected satellite-based rainfall estimates in simulating flash floods at Karpuz River - a semi-arid basin in Turkey. Global Satellite Mapping of Precipitation (GSMaP) product was evaluated with the rain gauge network at monthly and daily time-scales considering various time periods and rainfall rate thresholds. Literature lacks of studies using satellite rainfall estimates for flash flood modelling therefore the paper is relevant and of interest for the readers of the journal.

> **Response:** We thank Reviewer # 1 for his/her positive comments and support for our work.

**Comment 2:** Despite this, I think the paper contains serious shortcomings and its presentation is very poor. The main drawback is the analysis of the rainfall which is carried out on a monthly basis while flash flood occurrence time scale is often sub-daily and even sub-hourly (in this respect the authors claim in the abstract that the analysis has been carried out at daily time steps but no daily results can be found in the paper). An analysis at hourly and daily rainfall would be more appropriate for the study. The interpolation of rain gauges on the GSMaP grid seems wrong (I see many artefacts in Figures 15 and 22). An adequate discussion of the potential error of the interpolation (5 stations for obtaining rainfall at 0.1*0.1 degree on 2 x 1.5 degree area) should be present in the manuscript.

> **Response 2:** We thank Reviewer # 1 for these valuable comments. We will make sure to address the shortcomings and improve the presentation during the revision process. We concur with the statement that flash floods should be considered at sub-daily and hourly time scales in addition to monthly. We actually have already conducted the rainfall analysis at both daily and monthly time scales, but discussed only the monthly analysis in the manuscript to avoid excessive length of the manuscript. Thus, we will include the results and discussion of daily rainfall analysis during the revision process. Hourly analysis could not be performed due to lack of hourly rain-gauge observations; which is a common limitation for most of the semi-arid basins around the world. The interpolation methodology used is the Thiessen polygon method, and we believe that the most challenges in arid and semi-arid regions are the scarcity of the rain gauge network. Thus with this

study we are attempting to improve and correct the satellite rainfall data based on the (limited) available rain gauges to be used in flash floods simulation at the hourly time scale. We will double-check the interpolation procedure we have utilized. We will also make sure that the visualization of rainfall distribution maps are correctly processed in GIS.

**Comment 3:** The presentation of the paper lacks of an appropriate organization:
1. Intro section rationale should be:
A. Flash flood problems.
B. Use of gauge rainfall network problematic because there are too little number of Stations.
C. Possible alternative use of satellite data, problem with bias with satellite data,
D. Bias correction improves the hydrological model.

**Response 3:** We thank Reviewer # 1 for these valuable comments that would improve the flow of text and readability of the manuscript. We will surely take these comments into consideration during the revision process.

**Comment 4:** In the way it is presented it is difficult to follow.
**Comment 4.1**: Datasets description is totally missing (GSMaP is not described at all) and study area is described twice in section 2 and in section 4.1. after the result section. In section 4.1. no further information is given about the catchment characteristics neither about the discharge time series, event selection and so on. Here, only additional info about the flash flood problem are given (material that fits more for the introduction section).

**Response 4.1.** We thank Reviewer # 1 for these comments that will improve the organization of the manuscript. We will include the description of GSMaP in Section 2 (Study area and datasets). We will also introduce the study area in one section in more detail using a combined map.

**Comment 4.2:** The number of figures is enormous and redundant. Tables often contain the same information of the figures. 4. Performance scores are inadequate. NSE is used in rainfall assessment and not in flood assessment. I think it would be interesting to use categorical performance scores (Probability of detection and False alarm ratio) for rainfall assessment and RMSE and use NSE in the flood part. Based on that, I suggest the paper to be not acceptable and suggest to resubmit after being improved.

**Response 4.2.** Agreed. We will carefully revise the figures and the tables to remove any redundancy. We will also include categorical performance measures in the revised manuscript to improve the precipitation analysis.

**Comment 5:** I also have other comments that I will list below in order of appearance in the manuscript indicating also their relevance. The authors could take them into account for improving the manuscript. MODERATE: Pag. 3 lines 17-22 – Pag. 4 lines 1-8. This part should be moved at the beginning of the manuscript.

**Response 5:** Agreed. We will move indicated section to the beginning of the manuscript.

**Comment 6:** MODERATE: Pag. 2 lines 19-20. It seems the sentence is not a consequence of what is written before. Consider moving after describing potential problems of bias in satellite rainfall estimates.

**Response 6:** Agreed. We will re-write this section.

**Comment 7:** MINOR: Figure 1. Merge this figure with Figure 16.

**Response 7:** Agreed. We will merge these figures.

**Comment 8:** MAJOR: Pag. 4 lines 21-22. Why do you assess rainfall at monthly time scale then?

**Response 8:** As we discussed earlier and in the manuscript, raingauge networks are generally sparse and hourly observations are generally not available in semi-arid basins. Thus, we would like to make use of GSMaP product which is spatially continuous and available at hourly time scale for flash floods simulation. We have no other option but to asses GSMaP product at time scales for which rain gauge observations are available; daily and monthly. We aim to devise a correction procedure for GSMaP at these timescales and further apply this procedure at hourly timescale to investigate whether flood-simulations with a hydrological model improve pre- and post- bias correction procedure. Improvement in bias-corrected GSMaP driven hydrological model simulation as compared to flow observations provides an independent check on the performance of the devised bias correction procedure. We will make sure that the above message is clearly outlined in the revised manuscript. We have already conducted the monthly time scale based on daily data, therefore, it will be easy to include the daily analysis in the revised manuscript.

**Comment 9:** MAJOR. Pag 6 line 1- GSMap. Is not described in the text. Its description is relevant for the paper.

**Response 9:** Agreed. We will describe the GSMaP product in the relevant section.

**Comment 10:** MODERATE: Section 3. Explain better the difference between the PBIAS and BIAS and what information they should give one with respect to the other.

**Response 10**: Agreed. We will clarify the difference between these measures in the revised manuscript.

**Comment 11:** MODERATE: Pag. 8 lines 17-20. Not clear.

**Response 11: Agreed.** We will re-write and clarify this statement about the rainfall threshold selection during the revision process.

**Comment 12:** MINOR: figure 10 contains the same information of table 3. Consider removing.

**Response 12:** Agreed. We will avoid this redundancy by removing Figure 10.

**Comment 13:** MAJOR. Section 3.1. Point vs. grid comparison. Is not described in the methodology. What is the objective of this analysis?

**Response 13:** In this section, we aim to investigate the agreement between direct rain gauge observations (point-scale) with the overlying GSMaP grids. We will include a description of this procedure in the methodology section.

**Comment 14:** MODERATE. Pag 16 line 1-10. Please try to describe better this part. It seems very important for the paper.

**Response 14**: We will make every effort to improve and clarify the discussion provided in this section.

---

## Author Comment (AC2) · 10 Feb 2017

Dear Editor,

Please find below our detailed responses to Reviewer #2. We found the Reviewer comments very helpful and believe that the revisions detailed below will represent a significant improvement over the original submission in terms of both content and readability. In our responses below, we have clearly noted each comment and our specific response to those comments so that you can make a well-informed judgment. We would like to thank Reviewer #2 for his/her detailed comments on the paper.

Sincerely,

Mohamed Saber and Koray K. Yilmaz

**Overview Comment:** The work presents a study on the application of satellite-rainfall data for flash flood simulations for a basin in the Mediterranean region of Turkey. The first part of the work focuses on the analysis and bias correction of satellite rainfall estimates with reference to available rain gauges. The second part presents the hydrologic simulations based on corrected satellite-rainfall estimates. The topic of satellite-based flood simulation is of interest, particularly for data scarce regions as the one considered in this study. However, the work presented in this manuscript is limited in many aspects, which I highlight in more detail below, thus making it unsuitable for publication.

> **Response:** We thank the Reviewer # 2 his/her positive comments and support for our work. We think that the revisions based on the Reviewers' comments represent a major improvement over the original submission. The revisions detailed below improved both the content and the readability of the manuscript.

**Major comments**

**Comment 1:** Clarity and originality of objectives. It is not clear which is the major added value of this study. Is this the first work regarding satellite-rainfall and flash floods? I do not think so. Just a simple google search using the keywords (satellite, rainfall, flash floods) return several other past and recent works, which in fact were completely ignored from the authors. References need to be updated and authors need to contrast their objectives, methods and findings to what is already shown from previous studies. This will help the reader (and reviewer) to understand what is the added value of this work and subsequently how significant this is.

> **Response 1:** We thank the Reviewer for these valuable comments. During the revision process, we will make sure to highlight the added value of this work through detailed discussion of the relevant literature. We would like to note that majority of the earlier publications regarding the satellite-rainfall data and flash floods actually focused on humid regions within Europe, US, and Japan. On the other hand there exist only very few studies regarding semi-arid and arid regions especially along the Mediterranean region of the Middle East due to the scarcity of rain-gauge networks. To

the best of our knowledge this is the first study to compare/correct satellite-based rainfall datasets with raingauges along the Mediterranean region of Turkey.

**Comments 2:** Structure and coherence of manuscript. The coherence and structure of text in current version of the manuscript is very poor. At a large part I had the feeling that I was reading a document that was put together by simply pasting sections from a master thesis or similar. Many statements are unnecessarily repeated. The flow in developing the context in the introduction, but also other parts, is incomplete and sometimes certain statements are out of context. I provide some examples in the list of specific comments below.

> **Response 2:** We regret that the Reviewer pointed out the limitations on structure and coherence of the manuscript. We will make every effort to improve the flow of text and readability of the manuscript during the revision process.

**Comment 3:** Limitations in methodology and analysis. There is a general lack of clarity in many steps of the methodology that limits the understanding (at least of this reviewer). For example, was the bias adjustment factor derived and applied at monthly scale? Why not at daily scale? Also, was the spatial resolution of the hydrologic model at 1km? Do you think this is appropriate for modeling flash floods? What is the rational for selecting a rainfall threshold for the analysis? This is not explained well in the manuscript. Is the bias correction applied also for the values below the selected threshold (i.e. you may be increasing the "noise" as well if you do that). I provide more specific comments below.

> **Response 3:** We thank the Reviewer for these valuable comments that will help us improve the clarity of the methodological steps. We have actually performed the analysis and bias correction at both daily and monthly time scales, but we have not included the daily results to avoid excessive length. During the revision process, we will include the daily results as well. The hydrologic model is implemented at 1km grid-scale in accordance with the data availability and further requirements during model calibration. Our rationale in selecting the thresholds is based on the fact that satellite-based rainfall product reports many time steps with insignificant rainfall magnitudes. For example, upon comparing satellite-based vs. raingauge datasets for a threshold value of 00 mm (all the data were considered), we found that the number of rainy days was %50 more for satellite-based product compared to rain-gauge observations. We have thus checked different rainfall thresholds (1mm, 5mm, 10mm) and found that 1mm is the most appropriate threshold to be considered in the comparison process based on the statistical analysis. The bias correction procedure was applied to rainfall values above the selected threshold in an effort to reduce the data noise. We will include these details in the revised manuscript.

**Comment 4:** Limitations in findings and conclusions. According to the conclusions of the authors one of the important findings relate to the selection of an appropriate threshold for the correction of satellite. What is the rational for selecting this threshold? What are the potential dependencies of this threshold? This is important to generalize findings. But most importantly, several satellite-correction techniques have been proposed over the last decade that involve more advanced approaches (e.g. stochastic error models, distribution matching procedures etc) than the one proposed. Is the simple method applied in this study superior? If the authors wanted to demonstrate simply the effect of a simple correction on the hydrologic simulations, then they should have at least present the hydrologic analysis for both corrected and not corrected results.

> **Response 4:** We detailed the rationale for selecting a threshold in Response 3. Comparison of the performance of the proposed bias correction algorithm is out of the scope of this work. We, however, agree with the reviewer that including the flow simulation results before and after the

bias correction procedure will help to assess the value of the proposed bias correction procedure. We will include this analysis in the revised manuscript.

**Comment 5:** The authors state also that the "bias factors calculated in this study could be used for hydrological applications at any region with the same climatic conditions." Given the large differences of satellite-rainfall error properties reported from numerous evaluation studies, I am considering naïve to state that an adjustment factor could be unconditionally applied to another region of same climate…

**Response 5:** Agreed. We also think that our results could not be generalized without additional studies to confirm the findings. We will remove this statement.

**Comment 6:** Overall, I believe that more work is needed to bring the manuscript to a level adequate of publication and the work should focus on advancing the analysis as well as the writing/presentation of results.

**Response 6:** We thank the Reviewer for this valuable comment. We think that the revisions detailed in our responses to the Reviewers' comments significantly improves the originality and readability of the manuscript.

**Specific comments:**

1. L13-14 in the abstract is not clear. Please revise. In general the abstract is too descriptive and does not provide the overall objective of this work clearly.

   **Response**: Agreed. We will revise the abstract to clarify the objective and the results.

2. P2, L4: "hydrologic response is expected: : :.". It is not expected, it is proven by several studies.

   **Response**: Agreed.

3. P3, L1-4: Why you mention all these products when you are just using 1 of them? And by the way, why you considered only 1 product and did not include at least the high-res CMORPH?

   **Response:** We mentioned other satellite-based products to expose the reader to alternative datasets. Analysis including other satellite-based products (such as high resolution CMORPH and GPM) will be a logical extension of the present work as a future study.

4. P3,L8: ": : :model implementation and calibration: : :" please revise it is not clear.

   **Response:** Agreed. We will revise.

5. P3,L15-16: "..each flood acquires some particular and inherent characteristics of the occurrence locality". The meaning is totally unclear here.

   **Response:** Agreed. We will clarify the meaning of the sentence.

6. P4, L9-10: You mention the issue of model parameter uncertainty but how that relates to the rest of the context discussed in this work?

**Response:** We pointed out that the data scarcity is the main limitation to parameter identifiability in semi-arid regions. In this study, we utilized model parameter calibration to reduce (but not eliminate) the parameter uncertainty.

7. P8, L11: "underestimated bias" is not appropriate wording. Rainfall is underestimated not bias.
   **Response:** Agreed. We will correct the wording.

8. P8, L18-19: Interpretation provided is not meaningful. A rational for the selection of 1mm threshold is needed.

   **Response**: In accordance with our Response #3 we will detail the rationale for selecting 1mm threshold.

9. Why you present results on monthly scale (e.g. Fig2)? This temporal scale is completely our of flash flood context.

   **Response:** We will include the daily scale analysis in revised manuscript.

10. What is the difference between fig10 and fig 11?

    **Response:** We thank the Reviewer for this comment. Fig. 10 is for the total average over the time series (2007-2013), while Fig. 11 is for monthly average. We think retaining only one of these figures will be enough to highlight the differences between the thresholds and time series of dry and rainy seasons and the whole time periods. Thus we will remove one of these figures during revision.

11. P16, L16: ": : :tendency to underestimate rainfall: : :" is a result of NO DETECTION which is different from underestimation.

    **Response:** With this sentence we tried to describe "detection but with more underestimation comparing with the low land regions" which is also stated in many previous studies. We will revise the sentence to make sure the message is clear.

12. P17, L4-8: Is this the first time that these findings are reported?

    **Response:** According to the best of our knowledge, yes. We did not find any previous studies discussing this part in detail for semi-arid regions. But for other climatic regions there are many previous works. We will double check the literature to make sure that this statement is correct.

13. Section 3.2: Why repeating the procedure? Is bias adjustment factor from monthly comparison applied at hourly scale?

    **Response:** Unfortunately, there is no hourly raingage observations. Therefore we have utilized the monthly adjustment factor. During the revision process, we will also include the adjustment factors obtained from daily analysis for comparison and contrasting. The flow simulations pre and post bias correction procedure will serve as an independent check for the performance of the bias correction procedure.

14. Fig17 is not necessary as it is not related to main topic of this work. It is enough to provide reference to Ogzuler, 2003.

**Response:** Agreed. We will remove Figure 17.

15. P21, L7: I am not sure that "physically-based" is an appropriate description for this model. Do you consider the curve number and linear storage models as "physically-based" approaches?

    **Response:** Agreed. We will remove the "physically-based" description and describe the model as distributed hydrological model.

16. Fig19. Disaggregation (1km)". There was no mention on disaggregation of satellite rainfall estimates. If you mean that GSMaP rainfall was simply mapped on 1-km model pixels then this is not disaggregation.

    **Response:** The process we followed is to simply map the data over 1km model. We will revise the wording to avoid any confusion.

17. P25,L1: satellite rainfall data offer better "coverage" not "spatial resolution" as mentioned here.

    **Response:** Agreed. We will re-write this statement in line with the comment.

18. P28, L1-2: Explain how you identify areas prone to flash flood events?

    **Response:** We utilized the distribution maps and timing (time to peak) of simulated discharge to identify regions most prone to flash floods.

19. Fig.21 is simply a representation of the river network and in fact a coarse representation of the river network. It is not clear to me how this info can be used for "water management and disaster rist reduction" as stated by the authors.

    **Response:** The simulated discharge values obtained from this work could be used together with a detailed hydraulic model to identify inundated areas. We will revise the text to clarify this message.

20. P29, L1-2: "The main objective of the study is to enhance the capability of flash flood simulation using the corrected satellite-based rainfall data sets". You need to provide a comparison of hydrologic simulations between uncorrected and corrected. Also you hydrologic analysis should be focused on the flash flood events by providing error properties on their corresponding characteristics (e.g. flood peak estimation) instead of an overall efficiency metric that includes the whole time series.

    **Response:** Agreed. We will include a comparison of the hydrologic simulations driven by uncorrected and corrected rainfall. We will also include more hydrologically meaningful measures to investigate the performance of the model.